# Lockdown’s Silver Lining? Different Levels of Roadkill during the COVID-19 Times in Lithuania

**DOI:** 10.3390/ani13182918

**Published:** 2023-09-14

**Authors:** Linas Balčiauskas, Jos Stratford, Andrius Kučas, Laima Balčiauskienė

**Affiliations:** 1Nature Research Centre, Akademijos Str. 2, 08412 Vilnius, Lithuanialaima.balciauskiene@gamtc.lt (L.B.); 2European Commission, Joint Research Centre, Via Fermi 2749, 21027 Ispra, Italy

**Keywords:** COVID-19 lockdown, roadkill, domestic animals, wildlife, *Capreolus capreolus*, Lithuania

## Abstract

**Simple Summary:**

The impact of COVID-19-related human and vehicular mobility restrictions on the number of roadkills of wild mammals, and roe deer in particular, was assessed in Lithuania. Notably, there was a marked decrease in roadkill incidents on main, national and regional roads, particularly in April–May 2020 (the beginning of lockdown period I) and November–December 2020 (the beginning of lockdown period II). During these months, collisions with mammals on the A14 main road were reduced by 75–90%. However, there was a significant increase in the number of roadkills in urban areas, resulting in the total number of mammal–vehicle and roe deer–vehicle collisions in 2020 and 2021 being higher than expected based on long-term trends. However, after restrictions were eased, collision numbers with wildlife on the main, national and regional roads increased again and became higher than expected.

**Abstract:**

The impact of COVID-19 restrictions on human and vehicular mobility on the number of roadkills of wild mammals, and roe deer in particular, was assessed in Lithuania. We statistically analyzed the distribution of all mammals and roe deer (*Capreolus capreolus*) as the most abundant species annually from 2002 to 2022 and monthly from 2020 to 2021, including during the two restriction periods. Notably, there was a marked decrease in roadkill incidents on main, national and regional roads, particularly in April–May 2020 (the beginning of lockdown period I) and November–December 2020 (the beginning of lockdown period II), 5.1–20.8% and 33.6–54.4%, respectively. During these months, collisions with mammals on the A14 main road were reduced by 75–90%. However, there was a significant increase in the number of roadkills in urban areas, resulting in the total number of mammal–vehicle and roe deer–vehicle collisions in 2020 and 2021 being higher than expected based on long-term trends. However, after restrictions were eased, collision numbers with wildlife on the main, national and regional roads increased again and became higher than expected.

## 1. Introduction

Apparent from early on in the pandemic, there were multiple pathways via which the influence of COVID-19 has impacted societies, humans and wildlife, thereby highlighting complex relationships and the need for a holistic approach to their investigation [1]. Patterns of human–nature interactions, as well as human–wildlife interactions, were significantly affected by the global lockdown implemented to mitigate health risks [2]. Already at the beginning of the pandemic lockdown in March and April 2020, wildlife scientists took the opportunity to investigate wildlife responses [3], culminating in a Special Issue of the Conservation Biology journal in 2021, which covered a very wide range of countries and themes, and which provided a good starting point for further analyses [2].

The institutional consequences of the multifaceted COVID-19 pandemic have been largely negative, focusing on reduced environmental regulation and enforcement, weakened governance and reduced activity by conservation bodies [1,4]. Associated direct impacts on wildlife ranged from negative, such as reduced protection, to positive, such as reduced human mobility and the refuge effect of increasing access to suitable habitat areas [1]. However, many countries, including Lithuania, have still not carried out any analysis of their natural changes during this period.

In the United States, wildlife in protected areas responded positively to temporary closures [5], although management capacity was reduced due to staff reductions, changes in health and safety concerns and reduced revenues. The re-opening of these areas had negative impacts on species assemblages, including changes in animal activity and space use [6]. Among the negative impacts, reduced human disturbance associated with the closure of sites also benefited invasive alien species [7]. In Europe, Italy was the first country to implement closure measures [8], and it has also been a model for understanding the impact of lockdown on wildlife [7], concluding that the restriction of human activities had an overall positive impact on animal activity and distribution.

Despite closures, the conservation of biodiversity did not stop completely while, at the same time, benefits from reduced human activities were reported everywhere. There were sightings of wild species appearing in rural and urban areas where they had been absent for many years [9]. COVID-19 closures were shown to improve the quality and accessibility of urban and other anthropogenic habitats [10].

In COVID-19 pandemic areas in Canada and the US, it was reported that approximately 80% of observed bird populations “changed in pandemic-altered areas, usually increasing in comparison to pre-pandemic abundances in urban habitat, near major roads and airports, and in counties where lockdowns were more pronounced or occurred at the same time as peak bird migration” [11]. The number of wildlife–aircraft collisions at the 50 largest airports in the US declined between March and December 2020 [12].

It was, therefore, concluded that changes in human mobility had an impact on bird habitat use. These changes were not unidirectional but depended on a variety of habitat and bird behavioral responses [13]. For example, urban yellow-legged gulls (*Larus michahellis*) changed their diets in response to decreased human activity during the lockdown [14]. Another example was the increased body condition of greater snow geese (*Anser caerulescens atlanticus*) in 2020 due to changes in hunting practices imposed during the lockdown [15].

However, reductions in ground traffic were even stronger, and in some countries, they were so expressed that the new term “anthropause” emerged [16].

Decreased human presence during the pandemic inevitably resulted in lessened traffic loads, especially during lockdown phases. As a consequence, it was expected that roadkill numbers would drop [3]. This expectation was confirmed by Bíl et al. [17] in a study across 11 European countries. They found that the reduction in traffic led to a drop in wildlife mortality on the roads in four countries. In Sweden, where there was no lockdown, there was no observed reduction.

Many countries have already carried out roadkill analyses in relation to lockdowns [17,18,19,20,21,22,23,24,25,26,27,28]. However, Lithuania is not on this list. Furthermore, findings from various countries have shown inconsistency: while some reported no decrease in roadkill numbers on fenced highways [29] or in urban areas [30], others observed differing patterns of decrease [31], and some even experienced a brief decrease followed by an increase in roadkill numbers [26,32]. However, for mammals, the presence of humans may have only temporary effects [33].

Two outstanding publications provided a good starting point for roadkill analysis in an ecological context. Bruinderink and Hazebroek [34] pointed out that ungulate roadkills are not only a road safety issue in Europe but also impact population dynamics and sampling, with traffic intensity being an important factor, too. Extending the analysis to the US, Putmam [35] indicated that deer-related incidents have very high annual costs, and these are increasing. 

Roads have been shown to be important in conservation ecology [36], to have a negative influence on terrestrial and aquatic ecosystems [37] and to impact various components of landscapes [38]. Wildlife biologists were the first to observe the ecological consequences of roads, particularly through increased roadkills. Estimates have been alarming, showing that nearly 200 million birds and almost 30 million mammals are killed on the roads of Europe every year [39] and, at the global scale, threaten populations of eighty-three mammal species and even threaten four mammal species with extinction [40]. Recent estimates suggest that in extreme cases, up to 50% of mammal populations might be killed annually on roads, with roadkills accounting for as much as 80% of the species mortality [41].

Pandemic restrictions on human mobility, therefore, provided an unexpected experiment to understand the impact of transport on wildlife [42]. First, reports by citizen scientists showed a significant decrease in reported roadkills regardless of whether animals changed their mobility [19]. A similar observation was given by an insurance company in Lithuania, stating that the number of roadkills halved in April 2020 [43]. However, no scientific analysis followed.

Roe deer (*Capreolus capreolus*) was among the species most strongly impacted by lockdown in Slovenia [41]. This species appears to be capable of rapid changes in behavior, adjusting its activity patterns to anthropogenic load [44]. Therefore, we selected roe deer as the focal species for our analysis of lockdown influence, as it accounts for the highest number of roadkills among ungulates in Lithuania [45].

Reductions in roadkill rates during lockdown were observed in many countries. In the northwest of Spain, the maximum reduction was 64.8% during the most confined period in 2020, representing a 30.2% reduction compared to the same period in 2019 [24]. A similar figure, an 80% decrease in wildlife–vehicle collisions during the closure period, was observed in the UK [28]. In the US, roadkill decreased by 63–73% from mid-March to mid-April 2020 in four states [46]. In Poland, there was an over 50% decrease in hedgehog roadkills during the lockdown period [22], while Estonia, Spain, Israel and Czechia all experienced a more than 40% decrease in wildlife–vehicle collisions during the first weeks of lockdown [17]. South Korea saw a 19% reduction [2].

No significant changes in roadkill numbers were found in Sweden, with this country having the most liberal response to the pandemic [17]. Additionally, despite restrictions on human activity due to COVID-19 in Japan, decreased traffic volumes did not result in reduced roadkill numbers on the fenced highways [29]. In the US, the number of wildlife–vehicle collisions increased again as the pandemic gained momentum after a brief decline at the beginning [31]. In urban areas, the number of wildlife–vehicle collisions did not decrease, although it did decrease in suburban areas [30]. The changes in roadkill numbers were found to be associated with the type of road, as the traffic volumes and subsequent reductions due to restrictions varied between road types [16,24,46].

Possible explanations for these observed patterns include differences in accident reporting [5,19], changes in animal behavior and habitat use due to reduced human activities [18,19,32,47,48] and even gaps in previous knowledge about animal presence [49]. Disruptions in research activities and conservation enforcement may also obscure the true picture of these changes [9,50], as species react differently to disturbances [10,13], and this was also observed in the context of roadkill [44].

To comply with the above, our first research question was whether there was a reduction in Lithuanian roadkill rates of mammals, and especially of roe deer, during the periods of mobility restrictions in 2020 compared to the years 2019, 2021 and 2022. The second research question focused on the monthly patterns of roadkills during the months of mobility restrictions and whether they differed between different types of Lithuanian road networks (main, national and regional roads). We assumed that in March, April, November and December 2020, when the most restrictive lockdown measures were introduced, the number of roadkills should decrease the most. Additionally, we analyzed changes in road fatalities on the A14 main road, which is the most frequently covered by professional observers.

Our objective was to investigate the impact of human and vehicular mobility restrictions in Lithuania during the COVID-19 pandemic on the annual and monthly numbers of mammalian roadkills, including roe deer as the most representative species, and to assess if this impact varied across different road categories.

## 2. Materials and Methods

### 2.1. Lockdown Periods in Lithuania

Similar to most EU countries, Lithuania underwent two main periods of COVID-19 restrictions: the first in the first half of 2020 and the second from the end of 2020 until July 2021 (Table 1). During both restriction periods, there were phases where people were asked to refrain from leaving their residences as much as possible (though never legally prevented from) and, more so, were restricted from traveling between municipalities for non-essential purposes, except where they owned second properties in other municipalities. As such, transport movement between municipalities was controlled by the police but not absolutely forbidden. At the end of each restriction period, controls were loosened and restrictions eased. To reflect the peculiarities of Lithuania, we use the term “restriction” rather than shutdown or lockdown.

Between the first and second restriction periods, it was only recommended that persons restrict movement for non-essential reasons: people were asked to leave their place of residence only for work, essential shopping, medical reasons or other essential services, but transport movement was not controlled.

After the second restriction period, movement within the country became totally free; there were no curfews or movement restrictions.

We therefore define the first restriction period as March–June 2020 (four months) and the second as November 2020–June 2021 (eight months). Numerical estimations of traffic loads for the restriction periods are not available. However, information is available on the average annual daily traffic volume (AADT) for all road types [52]. On major roads, AADT was reduced by 8.5% in 2020 and by 3.2% in 2021 compared to 2019 levels. AADT decreased by 4.2% and 2.1% on national roads and by 7.9% and 1.0%, respectively, on regional roads (Table 2). We therefore expect to find that traffic volumes were much more reduced when there were COVID-19 restrictions on mobility. It is known that time spent driving following the implementation of restriction measures in Lithuania was reduced by 36% [2]. AADT on all roads in 2022 reached 2019 levels (Table 2).

### 2.2. Roadkill Data

We used data on roadkills in Lithuania for the period 2002–2022, limited to accidents involving wild and domestic mammals. The data sources for this study are the Lithuanian Police Traffic Surveillance Service and the Nature Research Centre. The first source provides information on police-recorded accidents involving mammals and vehicles, while the second source contains data on accidents recorded by professional theriologists driving in the country. Roadkill data included species, location and time. Species were mostly accurately recorded or, where not, were attributed to an “unknown mammal”. The details of data collection are discussed in detail in Balčiauskas et al. [45,50].

We analyzed 50,367 mammal–vehicle collisions (MVCs) between 2002 and 2022, of which 29,526 were roe deer–vehicle collisions (RDVCs), the single most frequent mammal recorded accounting for 19.1% to 74.9% of annual accidents (mean 58.1%, CI = 57.7–58.5%). The annual dynamics of MVCs and RDVCs are presented in Table 3.

All MVCs and RDVCs between 2002 and 2022 were categorized into either occurring on main, national, regional roads or other roads, with most of the latter category being in cities, towns, suburbs and other urbanized territories. Data on the length of the first three road types and the average annual daily traffic volumes are given by Balčiauskas et al. [50]. For the last group of roads, AADT data are not available, and the data coverage is limited to the period 2007–2022. Additionally, we analyzed MVCs and RDVCs on the main road A14, this being the most sampled road in Lithuania [45]. During the COVID-19 restriction months, 50 registration sessions were performed, each covering 62 km of road.

On the main roads, RDVCs accounted for 38.7% (CI = 37.9–39.5%) of all MVCs, with respective figures of 64.2% (CI = 63.5–64.9%) on national roads, 68.0% (CI = 67.0–69.1%) on regional roads and 68.6% (CI = 67.6–69.5%) on other roads. The sample sizes and annual dynamics of MVCs and RDVCs on different types of roads are provided in Appendix A.

### 2.3. Data Treatment

To assess the impact of the COVID-19 restrictions and the associated reduction in vehicle traffic, we compared the observed and predicted MVCs and RDVCs in the months when the restrictions were in place. Posterior predictive distributions of the expected monthly numbers were obtained from models made for every month of the COVID-19 restrictions for the periods of March–June 2020 and November 2020–June 2021. As the most conservative prediction, we used linear regression based on the least squares method.

The robustness of the differences between the observed and expected MVCs and RDVCs was assessed using chi-square statistics and the Wilcoxon signed-rank test (W) for equivalence of means. An insignificant test result proves the means of the observed and expected roadkills are equal. The proportions of RDVCs from MVCs were assessed as percentages, with Fisher’s 95% confidence intervals (CI). The lowest confidence level was set at *p* < 0.05, but at *p* < 0.10, the trend was not rejected. All calculations were carried out using PAST ver. 4.13 [53].

## 3. Results

### 3.1. Annual Dynamics of Mammal and Roe Deer Roadkill

On the main roads, a decrease in both MVCs and RDVCs was observed in both 2020 and 2021, the years with COVID-19 restrictions in Lithuania (Figure 1). The decrease in MVCs was 8.8% and 12.4% for the two years, totaling 220 fewer cases. The RDVCs decrease was smaller, 0.4% and 4.3%, respectively, totaling 25 fewer cases. However, neither decrease was found to be statistically significant (MVC: χ^2^ = 0.41; RDVC: χ^2^ = 0.25).

On national and regional roads, the trends differed, with a decrease in both MVCs and RDVCs in 2020, offset by an increase in collisions in 2021 (Figure 1). In numerical terms, there was a decrease in MVC cases on national roads by 5.5% in 2020, which amounted to 88 cases. However, this decrease was outweighed by an increase of 8.6% in 2021, equivalent to 145 additional cases. Regarding RDVC, there was a decrease of 6.4% in 2020, amounting to 72 cases, but this was countered by a significant increase of 12.9% in 2021, totaling 155 additional cases. Differences were significant (MVC χ^2^ = 7.95, *p* < 0.005; RDVC χ^2^ = 10.31, *p* < 0.002). On regional roads (Figure 1), MVCs decreased by 6.5% in 2020, i.e., 46 cases, and increased by 21.1% in 2021, i.e., 157 cases (χ^2^ = 12.41, *p* < 0.001). The decrease in RDVC in 2020 was 4.3%, equivalent to 23 cases, and the increase in 2021 was 24.1% or 135 cases (χ^2^ = 9.55, *p* < 0.005).

However, in general, the number of registered MVCs exceeded the expected figures based on long-term dynamics (8.4% increase in 2020, 14.9% increase in 2021, χ^2^ = 4.01, *p* < 0.05). Similarly, there was a noticeable trend of increase in the number of RDVCs (15.9% increase in 2020 and 23.5% increase in 2021, χ^2^ = 3.20, *p* = 0.07). This trend translates to nearly 1000 more roadkill incidents during the years with restrictions.

This increase depended on and was most pronounced on roads going through urbanized areas (see Figure 1). The number of registered MVCs exceeded the expected figures by 43.5% in 2020 and 31.7% in 2021, with a total increase of 880 collisions. However, this increase was not found to be statistically significant (χ^2^ = 2.46, *p* = 0.12). On the other hand, the number of registered RDVCs surpassed the expected figures by 800 collisions, showing an increase of 53.9% in 2020 and 35.5% in 2021, which was found to be statistically significant (χ^2^ = 4.36, *p* < 0.05).

### 3.2. Monthly Dynamics of Mammal and Roe Deer Roadkill during the COVID-19 Restrictions

In the months with restrictions, the number of observed MVCs was less than expected, with a difference of 232 cases on the main roads, 155 cases on the national roads and 29 cases on the regional roads (Table 4). Additionally, there were decreases of RDVCs by 73 on the main roads, 73 on the national roads and 7 on the regional roads (Table 5).

The most pronounced reduction in MVC during the period of restriction I was in April and May 2020; this was observed on main, national and regional roads, ranging from 5.1% to 20.8%. During the period of restriction II, the decrease in MVC was 33.6–54.4% in November and December 2020. In other months, the decline in road fatalities was not as consistent, with decreases in one category of road being outweighed by increases in another (Table 4).

As for the roe deer roadkills, a modest decrease in RDVC was observed during the period of restriction I on main, national and regional roads, with the reduction ranging from 5.9% to 17.1%. In the period of restriction II, the most pronounced reduction in RDVC occurred in December 2020, with 46.2–54.6% fewer roadkill cases than expected. However, in November 2020, the reduction in RDVC on main, national and regional roads, at 29.4–30.0%, was not sufficient to result in a negative balance for the total number of roadkills of this species. In the other months of both restriction periods, the pattern of roadkill number change was not consistent (see Table 5).

As examples of the most significant reductions, the spatial pattern of the MVC distribution in December 2019–2021 is depicted in Figure 2, while that of the RDVC distribution is shown in Figure 3. The presented maps reveal two main observations: firstly, roe deer accounted for the majority of roadkill incidents during the period of strict COVID-19 restrictions; secondly, there is a notable scarcity of accidents in proximity to major cities due to restricted travel between administrative regions. The administrative boundaries of core cities and towns (with populations of more than 20,000 inhabitants) are shown in Figure 2 and Figure 3, while labels show their names.

### 3.3. Monthly Dynamics of Mammal and Roe Deer Roadkill on the A14 Main Road during the COVID-19 Restriction Periods

Given the relatively low numbers of wildlife-related accidents on the A14 main road, the reductions in numbers during the periods of COVID-19 restrictions were very high in some months (see Figure 4). During the periods of restriction, the reduction of MVCs totaled 31 cases (χ^2^ = 25.62, *p* < 0.01). In the first restriction period, from March to June 2020, the decrease was 18.9–91.0% of the expected numbers, resulting in 38 fewer cases. However, in the second restriction period, the reductions in November 2020 and February 2021, seven and two cases, respectively, were outweighed by the other months.

As for roe deer, the total decrease during the restriction period was only three RDVC, which was not statistically significant (χ^2^ = 12.92, *p* = 0.30), and the pattern was not consistent (see Figure 4). The most notable decrease occurred from April to June 2020, with a reduction of up to 75.1% compared to expected numbers. Other significant reductions were observed in November and December 2020, with reductions of 83.1% and 63.5%, respectively. However, from January to May 2021, the observed number of RDVCs was higher than expected.

## 4. Discussion

We found that roadkill numbers decreased significantly on main, national and regional roads, especially in April–May 2020 (at the start of period I of lockdown) and November–December 2020 (at the start of period II of lockdown), from 5.1% to 20.8% and from 33.6% to 54.4%, respectively. In these specific months, the reduction of collisions with mammals on the main A14 road has gone from 75% to 90%. However, there has been a significant increase in the number of accidents in urban areas, so the total number of mammal–vehicle and roe deer–vehicle collisions in 2020 and 2021 is higher than what could be expected on the basis of the long-term trends. Nevertheless, the lifting of restrictions has led to a sharp increase in the number of collisions with wildlife on main, national and regional, again above the expected level.

Undoubtedly, data on roadkills allow the quantifying of road influence on biodiversity, particularly in higher vertebrates such as mammals and birds, but also, to a lesser extent, reptiles and amphibians [25]. Mammal roadkills have been found to be related to the day of the week and time of the day [24,54], but changes in animal activity due to climate changes and species abundance patterns should also be considered [55]. Therefore, in our analysis, we calculated regressions of expected roadkill numbers from data covering the period from 2002 to 2022.

In general, the decline in roadkill during the COVID-19 restrictions can be attributed to (1) reduced human activity, including reduced traffic volume, (2) changes in animal ecology and (3) changes in roadkill reporting. Our analysis focuses specifically on the effects of reduced human activity, based on the limitations of in-country travel (according to [51]), which, in some periods, even included physical blocking of roads [56]. We do not believe that reporting changes [4] had any influence on the patterns we identified, as we do not rely on citizen science for the reporting of roadkills [45,50].

According to published materials, increases in roadkills in urban areas during the COVID-19 period could be related to several factors, including changes in animal behavior [7,13,32], disrupted migration routes [18], encroachment of natural habitats [10], changes in food availability [10,48], decreased disturbance and fewer conflicts with humans [2,57] and reduced road maintenance [58]. We presume that the observed increase in the number of roadkills on the roads and streets of urbanized territories during the COVID-19 restrictions in Lithuania was a combination of different human mobility patterns (though Lithuania had no curfew, see [56]) and changes in animal behavior due to reduced stress [14,44,59]. We plan to analyze these roadkills further.

Our results can contribute to sustainable road development and maintenance (according to [40]) as they confirm that transport intensity is an essential factor affecting roadkill numbers. This relationship has been known long before the anthropause [49,60,61,62], and limitations during COVID-19 confirmed this through an experience that would otherwise have been unimaginable [16,47,63]. A decrease in human road deaths was also observed in many countries due to COVID-19 restrictions on mobility [64]. However, the effect of roadkill decrease in Lithuania was short-term, as the observed numbers were higher than expected in the months following the restrictions (see Table 4 and Table 5, Appendix A).

Changes in human–wildlife interactions during COVID-19 offered valuable insights into the potential impact of human activities on wildlife behavior and habitat use [65]. A strong example is China’s wildlife policy reform [66], which completely altered the relationship between animals and humans. The acceptability of wildlife was affected during the lockdown periods, and there may be long-term changes even after the COVID-19 pandemic [67], with consequences for both humans and nature [68]. In the future, not only citizen science but also emergency calls can be utilized to monitor human–wildlife interactions [69]. When these data are made openly available, scientists can use them for trend analyses. Considering that the effects of human influence on wildlife may vary depending on the species [70], it is crucial to analyze roadkill data for different species in Lithuania during the COVID-19-mediated anthropause, even if the sample sizes from that period are not as extensive as those for roe deer. By comparing similar data from other countries and accounting for the influence of transport intensity, researchers can quantify another challenge, namely climate change [57,71].

## 5. Conclusions

During the months of mobility restrictions due to the COVID-19 pandemic, the number of MVCs and RDVCs on the main, national and regional roads in Lithuanian was less than expected in comparison to the long-term trend, with the most pronounced reduction in April–May 2020 (beginning of lockdown period I) and November–December 2020 (beginning of lockdown period II);On A14, the main road with the best registration activity, MVC decreased by up to 90% and RDVC by up to 75% in comparison to the expected numbers in the same months;Collision numbers with wildlife on the main, national and regional roads exceeded the anticipated levels in the months immediately following the relaxation of restrictions on human mobility;The total numbers of MVCs and RDVCs in Lithuania in the years of the COVID-19 pandemic, 2020 and 2021, were higher than expected, according to long-term dynamics. Despite a marked decrease on the main, national and regional roads, this was due to the increased number of roadkills on roads in urban areas.

As further research, roadkill numbers in urbanized territories, that is, on roads other than main, national and regional roads, deserve special analysis. Two answers will be sought—roadkill diversity (list of killed species and their numbers) and the weekly or monthly influence of COVID-19 restrictions.

## Figures and Tables

**Figure 1 animals-13-02918-f001:**
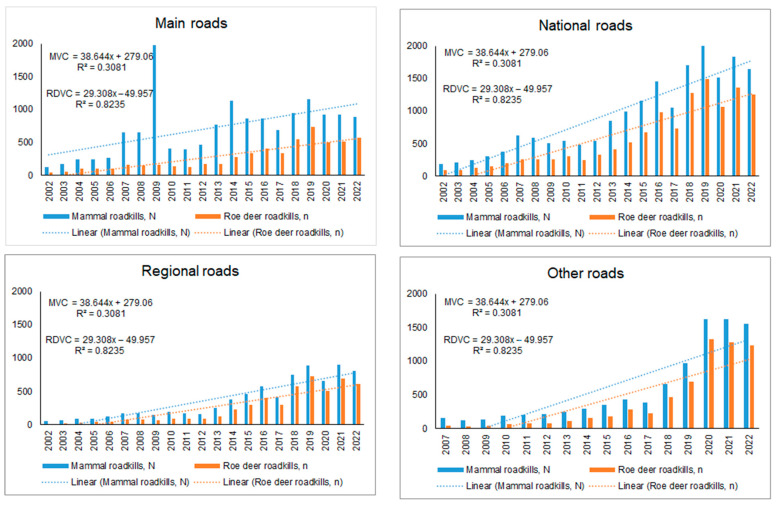
Mammal and roe deer roadkill dynamics on main, national, regional and other roads of Lithuania, 2002–2022. Regression lines represent expected annual roadkill numbers.

**Figure 2 animals-13-02918-f002:**
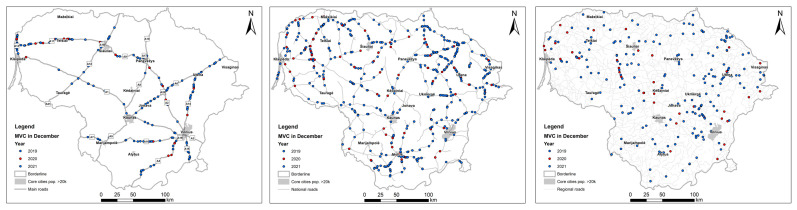
The pattern of mammal–vehicle collisions during restriction period II in December 2020 (shown in red) on the main, national and regional roads in Lithuania compared to mammalian roadkills in December 2019 and 2021 (shown in blue).

**Figure 3 animals-13-02918-f003:**
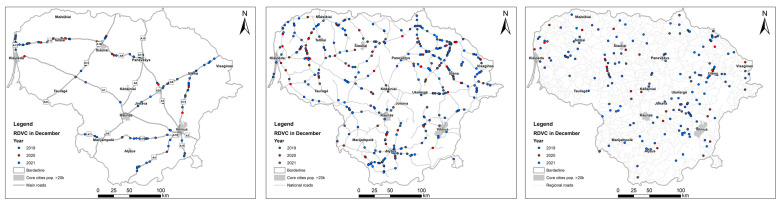
The pattern of roe deer–vehicle collisions during restriction period II in December 2020 (shown in red) on the main, national and regional roads in Lithuania compared to mammalian roadkills in December 2019 and 2021 (shown in blue).

**Figure 4 animals-13-02918-f004:**
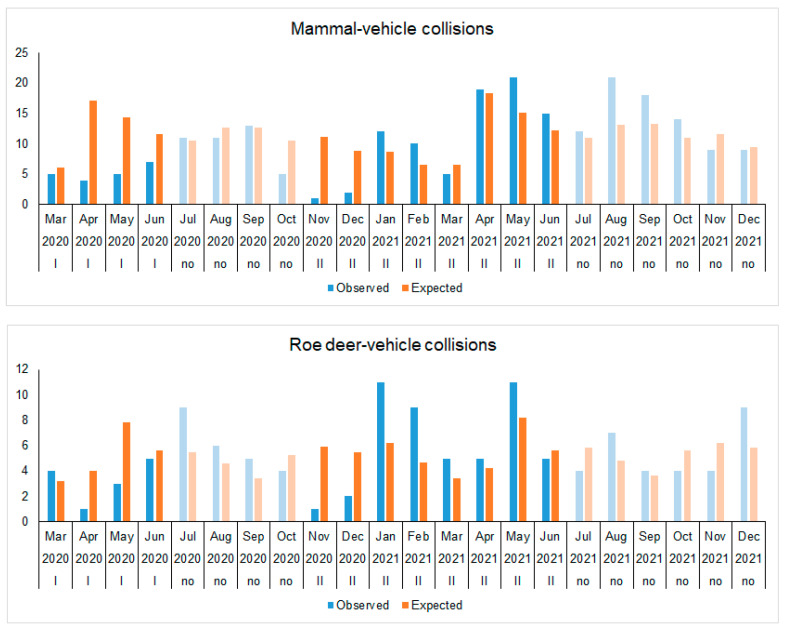
Mammal and roe deer roadkill dynamics in A14 main road of Lithuania, 2020–2021. Periods with unrestricted movements are shown in a paler color.

**Table 1 animals-13-02918-t001:** COVID-19 restriction periods in Lithuania, according to [51].

Restriction Period	Date	Length, Days (Months)	Transport Restrictions
I	26 February 2020–17 June 2020	112 (3.7)	Blocked
none	17 June 2020–4 November 2020	140 (4.7)	Limited
II	4 November 2020–2 July 2021	240 (8.0)	Blocked
none	2 July 2021–14 February 2022	227 (7.6)	Unlimited

**Table 2 animals-13-02918-t002:** Annual changes of average annual daily traffic volumes in Lithuania, 2010–2022.

Road Type	2010	2011	2012	2013	2014	2015	2016	2017	2018	2019	2020	2021	2022
Main	7268	7497	7518	7781	8274	8463	9047	9413	9614	10,010	9156	9692	10,260
National	1930	1944	1934	1978	2044	2109	2173	2261	2331	2381	2282	2330	2360
Regional	359	355	357	346	357	365	373	390	406	413	417	432	439

**Table 3 animals-13-02918-t003:** Sample sizes of mammal and roe deer roadkill data in Lithuania from 2002 to 2022.

Year	Total Mammals, N	Roe Deer, n	Roe Deer, %
2002	373	150	40.2
2003	447	167	37.4
2004	571	265	46.4
2005	641	295	46.0
2006	781	355	45.5
2007	1625	535	32.9
2008	1534	523	34.1
2009	2774	531	19.1
2010	1350	607	45.0
2011	1265	545	43.1
2012	1379	671	48.7
2013	2120	825	38.9
2014	2810	1198	42.6
2015	2824	1486	52.6
2016	3327	2075	62.4
2017	2554	1593	62.4
2018	4065	2870	70.6
2019	5027	3652	72.6
2020	4717	3393	71.9
2021	5281	3847	72.8
2022	4902	3673	74.9

**Table 4 animals-13-02918-t004:** Monthly dynamics of mammalian roadkills during the two periods of COVID-19 restrictions in Lithuania. Obs—observed number of roadkills; Exp—expected number of roadkills; Diff—difference in the number of cases (negative numbers indicate a decrease); %—difference in percentage from the expected number; decreases marked in color.

Period	Year	Month	All Roads				Main Roads				National Roads				Regional Roads			
			Obs	Exp	Diff	%	Obs	Exp	Diff	%	Obs	Exp	Diff	%	Obs	Exp	Diff	%
I	2020	Mar	229	254	−25	−9.8	46	44	2	3.5	90	97	−7	−7.3	42	46	−4	−7.8
I	2020	Apr	326	330	−4	−1.2	73	86	−13	−15.4	114	120	−6	−5.1	39	49	−10	−20.8
I	2020	May	590	535	55	10.2	155	180	−25	−13.8	147	171	−24	−13.9	64	71	−7	−9.4
I	2020	Jun	355	350	5	1.4	73	100	−27	−27.0	122	123	−1	−0.6	58	58	0	0.1
II	2020	Nov	452	460	−8	−1.7	40	76	−36	−47.5	115	173	−58	−33.6	50	80	−30	−37.8
II	2020	Dec	367	417	−50	−12.1	29	64	−35	−54.4	81	160	−79	−49.5	40	73	−33	−45.0
II	2021	Jan	370	326	44	13.6	41	49	−8	−16.6	111	130	−19	−14.5	54	56	−2	−3.8
II	2021	Feb	291	235	56	23.9	32	34	−2	−6.3	99	99	0	0.4	50	45	5	10.1
II	2021	Mar	332	269	63	23.3	34	46	−12	−26.8	110	103	7	7.2	72	48	24	48.5
II	2021	Apr	368	349	19	5.4	78	90	−12	−13.6	142	127	15	11.7	64	52	12	22.5
II	2021	May	542	566	−24	−4.2	137	188	−51	−27.2	191	180	11	6.0	81	75	6	8.0
II	2021	Jun	398	369	29	7.9	92	104	−12	−11.4	135	129	6	4.3	72	61	11	17.3
Chi-square test			χ^2^ = 21.96, *p* < 0.05				χ^2^ = 15.71, NS				χ^2^ = 35.10, *p* < 0.001				χ^2^ = 25.17, *p* < 0.01			
Wilcoxon test *			W = 54, NS				W = 76.5, *p* < 0.002				W = 46, NS				W = 35, NS			

*—if Wilcoxon test is not significant (NS), means are equal.

**Table 5 animals-13-02918-t005:** Monthly dynamics of roe deer roadkills during the two periods of COVID-19 restrictions in Lithuania. Obs—observed number of roadkills; Exp—expected number of roadkills; Diff—difference in the number of cases (negative numbers indicate a decrease); %—difference in percentage from the expected number; decreases marked in color.

Period	Year	Month	All Roads				Main Roads				National Roads				Regional Roads			
			Obs	Exp	Diff	%	Obs	Exp	Diff	%	Obs	Exp	Diff	%	Obs	Exp	Diff	%
I	2020	Mar	161	179	−18	−10.1	27	23	4	15.9	67	70	−3	−3.7	33	35	−2	−4.4
I	2020	Apr	238	227	11	5.0	41	49	−8	−17.1	80	85	−5	−5.9	34	37	−3	−7.6
I	2020	May	487	396	91	22.9	122	116	6	5.2	117	134	−17	−12.5	52	58	−6	−10.2
I	2020	Jun	253	229	24	10.3	40	52	−12	−22.8	90	89	1	1.0	41	42	−1	−3.0
II	2020	Nov	371	328	43	13.0	28	40	−12	−30.0	87	124	−37	−30.0	43	61	−18	−29.4
II	2020	Dec	299	305	−6	−2.0	17	37	−20	−54.6	64	119	−55	−46.2	31	59	−28	−47.4
II	2021	Jan	310	244	66	27.0	33	29	4	12.2	96	99	−3	−3.0	42	44	−2	−5.3
II	2021	Feb	222	185	37	20.0	26	23	3	15.4	72	75	−3	−3.8	37	34	3	9.2
II	2021	Mar	269	191	78	40.8	26	25	1	5.3	88	74	14	19.1	55	37	18	49.1
II	2021	Apr	256	241	15	6.1	39	52	−13	−25.5	100	90	10	10.9	47	39	8	20.0
II	2021	May	438	421	17	4.0	96	123	−27	−21.7	159	142	17	12.1	76	62	14	23.1
II	2021	Jun	273	244	29	12.0	50	55	−5	−8.6	102	94	8	8.1	54	45	9	20.1
Chi-square test			χ^2^ = 19.30, *p* = 0.09				χ^2^ = 12.88, NS				χ^2^ = 25.25, *p* < 0.01				χ^2^ = 18.97, *p* = 0.06			
Wilcoxon test *			W = 72, *p* < 0.01				W = 62, *p* = 0.07				W = 46.5, NS				W = 39, NS			

*—if Wilcoxon test is not significant (NS), both means are equal.

## Data Availability

The entire dataset cannot be published in an open access system, as the data may contain sensitive information. However, the numerical values and patterns of roadkills used in this paper are provided in the Appendix A.

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
