# Peer review of "Lockdown’s Silver Lining? Different Levels of Roadkill during the COVID-19 Times in Lithuania"

_animals, 2023, doi:10.3390/ani13182918_

Round 1

Reviewer 1 Report

Comments to authors

Lockdown’s silver lining? Different levels of roadkill during COVID times in Lithuania

This study revealed the change of roadkill during periods of human activity restriction in Lithuania.

Roadkill decreasing or not decreasing are shown by some papers. This report in Lithuania is necessary to prevent roadkill problem.

However, I have some comments to this paper. I attached a file of comments.

Author Response

Rev#1 Comments to authors

Lockdown’s silver lining? Different levels of roadkill during COVID times in Lithuania

This study revealed the change of roadkill during periods of human activity restriction in Lithuania. Roadkill decreasing or not decreasing are shown by some papers. This report in Lithuania is necessary to prevent roadkill problem.

However, I have some comments to this paper.

Comment 1. Some references are wrong, for example, line 94 (33?) and line 109 (48?).

Answer: we apologize both mistakes, the first one was just mistype, the second one, unfortunately, was our oversight and resulted the number changes of the references.

Comment 2. I understand that AADT data is nothing during restriction period, however, traffic trend during other periods should be shown due to extremely important to understand roadkill dynamics. I think that authors should show traffic volume trend at least one road. Recovery after restriction period also should be written.

Answer: as we already wrote, unfortunately, monthly numbers of AADT are not available. To acknowledge your comment, we present decrease of annual AADT, based on data from authorities (https://lakd.lt/eismo-intensyvumas) as new Table 2. We also added text to this new table, about percent annual AADT decreases on the three types of roads.

Comment 3. Result (3.1)

-Line225: It was indicated that MVCs and RDVCs decreased in both 2020 and 2021. Figure 1 might show the rapidly increase in 2018 and 2019. Are MVCs and RDVCs decreasing apparent decrease?

Answer: roadkill numbers were steadily increasing, so there were no factors, apart traffic limitation during COVID, to influence roadkill number drop in 2020 and 2021 on the main, national and regional roads. We added information as Table 2, to show AADT decrease in these years.

-MVCs and RDVCs are compared by real value. However, number of roadkill gradually increase. I guess that this increase is to be the increase of road extension. I suggest that authors compare by using MVCs and RDVCs reflected road extension in each year.

Answer: based on the open data from the Lithuanian Road Administration, road length in the categories of main, national and regional roads is relatively stable, that is, new roads were not building in the analysed period of time, and extension was less than 1%. Therefore, we do not see this as a possible factor of roadkill.

-Expected value was calculated using linear regression. I think that linear regression could not reflect trend of month and year. Should use time series analysis?

Answer: using least squares, we tested not only linear regression, but also other models and selected one with the best fit. For sure, using time series with/or polynomial fitting, it is possible to get better fit to empiric data of roadkill. However, these fits are poorly interpreted. Linear regression is to show that roadkills were steadily increasing, except of the years with COVIF limitations (2020 and 2021), and in the months when limitations were in place. This decrease was statistically significant.

When regression is better than the time series?

Time series is more suitable for forecasting and detecting patterns in temporal data, while regression is more suitable for estimating and explaining the effect of variables on an outcome. We do not make any prognosis here, as 2020 and 2021 were backward from 2022.

Consulting with web resources on statistical analysis, such as https://stackexchange.com/, https://data.ucsf.edu/research/stat-analysis, https://www.statisticssolutions.com/free-resources/directory-of-statistical-analyses/#regression-analysis, https://biostats4you.umn.edu/, and other, we decided to use regression as the most conservative assessment of expected roadkill numbers. It was said, that in practice, really advanced models do well on in-sample forecasts but not so great out in the wild, as compared to more simpler models.

We do not analyze or model the temporal patterns, trends, and dependencies within the data. Our purpose was very simple – to say, if observed roadkills are in line with prediction of the number growth, as found from long-term data.

Comment 4. P3

I could not distinguish colors and points in figure2 and figure 3.

Answer: unfortunately, it was a consequence of scale, to fit three maps across the page. We made symbols larger, now these are visible. Grey changed to blue for better contrast.

Comment 5. Discussion

Results were not enough shown in discussion. Authors should carefully discuss to each result.

Answer: we extended Discussion section with a short summary of findings in the beginning. However, it should be mentioned again, that we couldn’t relate monthly roadkill to AADT, as monthly AADT are not available in Lithuania.

Reviewer 2 Report

Authors are kindly asked to include in the keywords: Capreolus capreolus.

Figures 2 and 3 are not very clear and legible, furthermore the colors, in particular the red, are not appreciated well. I don't know if this is due to the level of definition but it would be good to redraw both these figures more clearly, with more evident colors and larger dimensions.

Author Response

Rev #2 comments

Comment: Authors are kindly asked to include in the keywords: Capreolus capreolus.

Answer: done, thank you for reminding this.

Comment: Figures 2 and 3 are not very clear and legible, furthermore the colors, in particular the red, are not appreciated well. I don't know if this is due to the level of definition but it would be good to redraw both these figures more clearly, with more evident colors and larger dimensions.

Answer: unfortunately, it was a consequence of scale, to fit three maps across the page. We made symbols larger, now these are visible. Grey changed to blue for better contrast.

Round 2

Reviewer 1 Report

Comments to authors

Authors replied to my question and comments.

I have only one question about the decreasing of AADT.

Authors added AADT data in Table 2, I think that these values of 2020 and 2021 may not decrease compared with 2019. Reduced percentage is also low. For example, Do AADT fluctuate among years as 2019-2022? Could you show AADT data of more long term?

Author Response

please find answer and data attached
